# Processed Foods and Nutrition Transition in the Pacific: Regional Trends, Patterns and Food System Drivers

**DOI:** 10.3390/nu11061328

**Published:** 2019-06-13

**Authors:** Katherine Sievert, Mark Lawrence, Asaeli Naika, Phillip Baker

**Affiliations:** 1School of Exercise and Nutrition Science, Deakin University, Geelong VIC 3220, Australia; 2Institute for Physical Activity and Nutrition, School of Exercise and Nutrition Science, Deakin University, Geelong VIC 3220, Australia; mark.lawrence@deakin.edu.au (M.L.); phil.baker@deakin.edu.au (P.B.); 3National Food and Nutrition Centre, Suva, Fiji; asaelinaika@gmail.com

**Keywords:** Pacific Islands, processed foods, nutrition transition, noncommunicable diseases

## Abstract

Background: The role of processed foods in nutrition transition in the Pacific is receiving some attention in the context of a significant obesity and diet-related noncommunicable disease health burden. However, trends, patterns and underlying drivers of processed food markets in the Pacific are not well understood. The aim of this study was to investigate recent trends and patterns of processed food markets in the region and interpret the findings by engaging key literature on relevant food systems drivers. Methods: We conducted a mixed-methods approach involving two steps; (1) We analysed Euromonitor market sales data for processed food and beverage products sold from 2004–2018 for 16 countries differentiated by income level, and (2) guided by a food systems conceptual framework, we drew upon key literature to understand the likely drivers of our observations. Results: We observed plateaus and declines in processed food sales in some high-income countries but increases in upper-middle and lower-middle income countries, and most rapidly in the latter. Beverage markets appear to be stagnating across all income groups. Carbonated soft drinks, baked goods, vegetable oils, processed meats, noodles and sweet biscuits made up the majority of sales in transitioning countries. These observations are likely a result of income growth, urbanising populations, trade and globalisation, and various policies implemented by Pacific governments. Conclusions: A processed foods nutrition transition is well underway in the Pacific region and accelerating most prominently in lower-middle income countries.

## 1. Introduction

Unhealthy diets and poor nutrition are now the leading contributors to the global burden of disease [1]. Nowhere is this more so than in the Pacific region, where overweight, obesity and diet-related noncommunicable diseases (NCDs) exact a substantial social and economic toll. According to the World Health Organization (WHO), eight of the world’s ten most obese nations are Pacific Island Countries (PICs) [2], with 33% of all adults in the region living with overweight or obesity in 2014 [3]. At the regional level, mean body mass index (BMI) has risen by ~2.0 kg/m^2^ per decade between 1980 and 2008, five times the mean increase worldwide [4]. NCDs are responsible for around 70% of all deaths in PICs, and as a result, life expectancy has fallen in some countries [4]. The picture is similar among the sizeable Pacific populations (i.e., diaspora) residing in the high-income countries of Australia, New Zealand and the United States (US) [5]. Such extensive levels of diet-related ill-health both cause and exacerbate poverty, impeding regional economic and social development [6]. The costs of diet-related NCDs to households include loss of employment opportunities (for both individuals and carers) and financial burdens of living with disability, particularly in countries without universal healthcare [7]. Some countries are also experiencing a double burden of malnutrition. For example, middle-income countries like Fiji and Tuvalu have high levels of both overweight/obesity and child stunting [8]. 

Against this backdrop, the role of processed foods in the diet-related health of Pacific populations has been the subject of investigation previously [9,10,11,12] and deserves attention for several reasons. We define processed food products as substances extracted and refined from unprocessed or minimally processed foods that are ‘ready-to-eat’ or ‘ready-to-heat’, made from industrially prepared ingredients and additives, usually highly palatable and intensively marketed, and often high in free sugars, trans-fats and low in micronutrients [13]. There are several frameworks used to categorise foods by their level of processing [14,15,16]. Most of the food products included in this analysis are considered ultra- or highly-processed in these frameworks. However, we also included some culinary ingredients (e.g., vegetable oils) and basic processed food (e.g., bread) product categories, which have their own additional critical implications for nutrition in transitioning countries [17,18]. 

The processing of foods, as practised by humans for millennia, is not in and of itself detrimental for health. Advances in food processing technologies have helped to improve worldwide food security by reducing the perishability of foods [19]. However, processed foods and especially ultra-processed foods (UPFs) contribute to long-term adverse health outcomes [1,20,21,22]. A growing body of evidence links a higher proportion of UPFs in the diet with obesity, cancer, cardiovascular disease, metabolic risks and all-cause mortality [23,24,25,26]. Whilst understanding of the exact mechanisms linking food processing with health outcomes is still developing, some plausible hypotheses are emerging [22]. First, such foods are often energy-dense and contain high quantities of ‘risk’ nutrients, namely free sugars, sodium and trans-fats [26]. Second, they can displace whole and minimally-processed foods, thereby reducing total dietary quality [27,28]. Third, varying methods of industrial processing itself can affect the physical structure and chemical composition of the food, which can have an adverse impact on metabolic processes. For instance, processing can result in modified structural properties of the food that lead to higher glycaemic responses and lower satiety signals [29,30]; and the use of certain industrial ingredients may result in gut microflora dysbiosis, increased gut permeability and inflammation [31,32]. 

Recent analyses have also started to unpack the links between food systems change, processed foods and the nutrition transition [33,34], with some focused on the Pacific in particular due to its disproportionately high NCD burden [9,12]. Transformations underway in food systems—inclusive of food supply chains, food environments, consumer behaviour and the regulatory frameworks shaping those systems—powerfully shape food choices and hence population dietary patterns [35,36]. The theory of nutrition transition proposes that with urbanisation, rising incomes, labour market and technological change, there is a shift away from traditional diets to those higher in vegetable oils, sweeteners and refined carbohydrates, and animal-sourced foods [37,38]. Many empirical studies now support this theory, with later studies implicating increased processed food consumption as a key feature of the nutrition transition [18,39,40]. 

The nutrition transition not only involves a shift from more traditional to globalised and processed foods, but also substantial changes in the way people source and prepare food linked with convenience, price and changing culinary practices [41]. A growing number of studies also emphasise the ‘political economy’ drivers of food systems, including trade and investment liberalisation, the increasing penetration of transnational food companies into the Global South and inadequate regulatory frameworks to promote and protect public health in these new contexts [40,42,43]. At the same time, because these food systems forces have varying and context-dependent impacts on the availability, affordability and desirability of different food types, there is no singular or uniform nutrition transition. There may be a convergence in the global diet towards the consumption of more processed foods in general, but divergences resulting from variegated food systems drivers at regional, national and local levels [18,39]. 

Despite the likely importance of processed foods to the nutrition transition and public health in the Pacific, few analyses of regional trends, patterns and food systems drivers of processed food markets exist. Although some previous literature reported on the role of processed foods in Pacific diets, no regional country comparison exists, nor do these studies address food systems drivers. In this analysis, we aimed to address this gap by describing changes in PIC processed food markets as well as in countries with large Pacific Island diaspora and engage with the literature on potential food system drivers. We addressed several key questions. First, to what extent has growth in processed food markets continued, accelerated or abated in Pacific countries? Second, are all countries undergoing the same ‘processed food transition’, or are there categorical differences between them? Third, what are the likely food system drivers behind the observed trends? As PICs have been relatively progressive in adopting regulations, taxes and programs in order to combat their significant diet-related health burdens [44], we also considered the role of policies in shaping processed food markets.

## 2. Materials and Methods 

To investigate recent trends, patterns and drivers of processed food markets across the Pacific, we adopted a mixed-methods approach involving two steps. First, using market sales data for food and beverages sold between 2004 and 2018, we conducted a quantitative analysis of the apparent consumption (i.e., formal sales) of processed food and beverage product categories. Second, guided by a food systems conceptual framework [35], we drew upon key literature to identify potential drivers of the observed trends and patterns.

### 2.1. Countries

We included 16 countries of the Pacific Rim and Pacific Ocean (henceforth, the ‘Pacific’) in the analysis, for which market data were available. This included the USA, Australia and New Zealand as countries with large Pacific Island populations (i.e., diaspora). Countries were organised by income category according to standard World Bank lending classifications [45]: Lower-middle income countries (L-MIC) ($996–$3,895) were Kiribati, Papua New Guinea, Solomon Islands and Vanuatu; upper-middle income countries (U-MIC) ($3,896–$12,055) American Samoa, Fiji, Nauru, Samoa, Tonga and Tuvalu; and high-income countries (H-IC) ($12,056 or more) Australia, French Polynesia, Guam, New Caledonia, New Zealand and the US.

### 2.2. Data Sources, Product and Ingredients Categorisation

Sales volume data for processed food and beverage product categories (Table 1) were sourced from the Euromonitor International (Euromonitor) Passport Global Market Information Database, 2019 Edition. These data were retrieved by Euromonitor from various sources, including trade associations, industry bodies, business press, company financial reports and official government statistics. Data were then validated by individuals working within the food industry. Product categories are defined by Euromonitor [46] and are aggregations of both processed and ultra-processed foods. These Euromonitor packaged food categories have been used in similar analyses of processed food consumption markets [39,47]. Given that vegetable oils are noted to be a significant element of the nutrition transition [48], we employed a broader definition of “processed foods” as substances extracted and purified from unprocessed or minimally processed foods (e.g., vegetable oils) and industrially produced ready-to-eat or ready-to-heat food products resulting from the processing of several food substances (e.g., sweet biscuits) [39]. Data on total per capita sales volumes (kg per capita) of processed food products (through retail and food service outlets), from 2004–2018, were extracted for the categories in Table 1. 

Data for American Samoa, Fiji, French Polynesia, Guam, Kiribati, Nauru, New Caledonia, Papua New Guinea, Samoa, Solomon Islands, Tonga, Tuvalu and Vanuatu were modelled data. According to Euromonitor, this modelling involves allocating three to four relevant approximator countries to generate initial estimates, with parity ratios determined from a set of economic and demographic factors. When constructing this similarity index, some 50 indicators are assessed, including GDP per capita and real GDP growth rate, urbanisation rate, average household size, number of children, birth rates, age distribution indicators, infant mortality, life expectancy, adult literacy rate, corruption perceptions indices, female employment rates, religion and ethnicity indicators, climate indicators, trade intensity and geographical distance between countries. Similarities are then compared for all of these indicators across every possible pairing of countries. Further adjustments were made where real country-level sales data were available. These estimates were then checked by industry experts [46,49]. 

## 3. Results

The determination of trends and country-level variations in per capita processed food and beverage category sales provided three powerful observations. 

Observation 1: Processed food sales are increasing in U-MICs and L-MICs, and most rapidly in the latter, albeit from a low baseline. Figure 1A–C shows volumes of processed foods sales per capita by country income group. In almost all countries, processed food sales increased from 2004–2018, most notably in U-MIC and L-MICs. Sales of baked goods and edible oils in particular were higher than most other categories, particularly in U-MICs and L-MICs. In H-ICs, processed foods sales were increasing in Australia and New Caledonia but plateauing or even declining in the others. In the US, for example, sales declined by 6.2% over the period. By contrast, sales increased in all U-MICs and L-MICs. Among U-MICs, Nauru and Fiji showed the greatest increase from 46.4 to 52.7 (14%) and 25.0 to 32.1 (28%) kg per capita, respectively. Sales growth was most rapid in L-MICs, with all countries increasing by at least 40% over the period, although with absolute volumes well below H-ICs. By 2018, Vanuatu was displaying volumes akin to those of U-MICs, with sales of 19.5 kg per capita. Papua New Guinea displayed the highest level of growth, increasing from 7.7 kg/p/cap in 2004 to 12.0 kg/p/cap in 2018 (56%). 

Important to note is that overall volumes in U-MICs and L-MICs are much lower than in H-ICs, with absolute amounts reaching close to 100–190 kg per capita in H-ICs compared with only up to 20 kg per capita in L-MICs. This may be reflective of limitations in the data as Euromonitor can only capture formal channels of food sales, and as such may be omitting less formal outlets such as street vendors. Within these groups, there were also varying patterns. Firstly, edible oils have a much larger proportion of sales volumes in U-MICs (second highest volume) and L-MICs (highest volume) than in H-ICs (eighth highest volume). Furthermore, products that require refrigeration such as ice cream and cheeses feature more prominently in H-ICs than in L-MICs and U-MICs, whereas the less perishable and inexpensive categories of noodles and sweet biscuits have higher volumes in the latter. Processed meat features prominently in all income groups, with the second largest volume in H-ICs and the fourth largest in U-MICs and L-MICs. 

Observation 2: Beverage sales are increasing in U-MICs and L-MICs, most rapidly in the former, with sales in some countries approaching H-IC levels. Figure 2 demonstrates beverage sales volumes per capita by country income group. Like processed foods, the scale of absolute volumes differed across income groups. H-ICs such as the US reached over 250 kg per capita in soft drinks sales in 2004, whereas L-MICs such as Vanuatu were one fifth of this amount, with almost 50 kg per capita in sales by 2018. Also similar to the findings on processed foods, total beverage sales volumes increased the most in U-MICs and L-MICs, with sales plateauing or in some cases declining in H-ICs. The most notable product category increases in U-MICs and L-MICs were carbonated soft drinks and juice. Similar to the data for processed foods, the US saw decreased sales across all soft drink categories, from 270.5 litres per capita to 223.2 litres per capita (17% decrease). As with processed foods, US sales far exceeded those of all other countries across all income levels. Whilst at a lower scale, beverage sales increased in almost all U-MICs and L-MICs, and most notably in the former. For example, Samoa increased apparent consumption of carbonates from 42.5 to 46.6 litres per capita in 2004–2018. In this same year, American Samoa saw sales of carbonates decline from 71.2 to 68.3 litres per capita, while juice sales increased from 13.3 to 18.8 litres per capita. Tuvalu and Nauru also had increases in juice sales from 6.4 to 11.1 (73%) and 8.4 to 15.3 (82%) litres per capita, respectively. 

Observation 3: Vegetable oil sales, especially palm and coconut, are increasing rapidly in U-MICs and L-MICs. Per capita sales of vegetable oils, particularly palm and other categories, including coconut and blended vegetable oil, is increasing rapidly in U-MICs and L-MICs (Figure 3). This is consistent with existing literature indicating that vegetable oils (including coconut oil) are now providing the largest proportion of fat to PIC diets [12]. Furthermore, by 2018, U-MIC countries like American Samoa and Nauru had per capita sales volumes of around double that of high-income countries like Australia and New Zealand (8.3, 6.1 vs. 3.9, 2.9 litres/cap respectively). Furthermore, palm oil was noted to form a distinct proportion of the sales in U-MICs, for example, exceeding all other oil varieties in Fiji (4.5 litres of palm oil vs. 2.7 other oils in 2018).

## 4. Discussion

This paper has endeavoured to answer several questions. Are processed foods markets expanding or contracting in the Pacific region? If so, how do changes vary across countries at different stages of economic and social transition? And finally, what might explain the observed trends and patterns in processed food and beverage markets in the Pacific? To answer these questions, we drew upon the results and further engaged with key literature on the food systems drivers of diets and nutrition, with reference to PICs in particular. 

The findings of this study are largely consistent with nutrition transition theory. We demonstrated that processed foods play a prominent and increasing role in the diets of Pacific populations, particularly in U-MICs and L-MICs undergoing economic and social transition. By contrast, the apparent consumption of processed foods and beverages in some H-ICs is plateauing and, in some cases, even declining. Several processed food categories appear to play a greater role than others in the nutrition in the Pacific, in particular, baked goods, vegetable oils, processed meat, noodles, sweet biscuits and carbonated soft drinks. Juice, which is often high in free-sugars, may also be playing a more prominent role in people’s diets in some U-MICs.

### 4.1. What Factors Are Driving a Processed Food Transition in Pacific Countries?

Processed food consumption is not solely the end-product of individual-level choices. The availability, affordability and desirability of foods consumed is powerfully shaped by the surrounding food system and an array of interrelated and variable drivers spanning global to local scales. In the Pacific specifically, the rise in household incomes, urbanisation, and participation of women in the workforce have positioned processed foods as both convenient and a marker of social status [50,51]. Coupled with this, the participation of PICs in global free trade agreements as well as market penetration of transnational food and beverage corporations (TNFBC) has made processed foods more widely available, even in very remote localities [4]. This has resulted in a health burden in the region, leading to various frameworks being launched, aimed at addressing this NCD burden [52].

The past forty years in particular have witnessed sweeping changes in the way people in the Pacific buy and consume food [53]. It is predicted that by 2020, more than half of all Pacific Islanders will be living in urban areas [54]. This mostly stems from a desire to access better education and employment opportunities, health services, and to adopt a modern lifestyle [55]. As progressively urbanised lifestyles face restraints on time and available cooking space, much of the food and beverage demand favours those that can be consumed with convenience. This means that food products that can be stored for long periods of time without perishing—i.e., processed foods—have become increasingly present in the region. Urbanisation also correlates with increased intake of saturated fats, sugar, and other refined carbohydrates, ingredients that processed foods mostly comprise [56,57]. 

Our findings of the growth in sales of vegetable oils in the region is likely indicative of trends towards urban living that have not been conducive to traditional styles of cooking, due to the limitations of time and physical space required. Our observed rise in vegetable oil sales is likely suggestive of the movement away from a largely fat-free method of cooking underground, i.e., “earth oven”, towards frying and deep-frying foods (such as chicken), which are then accompanied by rice and/or bread that are often also cooked with oil [12]. These practices are likely more convenient than traditional methods, which may increasingly be used for special occasions only. The increase in vegetable oils and especially palm oil sales in many countries appears to be a noteworthy feature of the nutrition transition in the Pacific. Others have reported that as countries transition, vegetable oils appear to markedly increase in availability and consumption as a cheap and convenient culinary ingredient used in home cooking. Popkin notes that while rapid increases in edible oils have been a significant element of the nutrition transition across countries of all levels of income, the net dietary impact is comparatively greater in low-income countries [48]. We note that there is increasing recognition of the risks this category poses to overnutrition in the region, for example, with Fiji advising limited consumption in its dietary guidelines.

The variation in urbanisation may partly explain our observed disparities in absolute amounts of sales between H-ICs and U-MICs/L-MICs and explain why growth of processed foods is not taking place at the same rate as in South East Asian countries at a comparable stage of economic transition [39]. In many PICs, the proportion of the population living in urbanised areas, and therefore with more immediate access to, and capacity to buy, processed foods, is considerably smaller than those living in rural areas. For example, only 18% of Samoans and 23.1% of Tongans are living in urban areas [58]. However, this does not provide a complete account of our observations. Countries such as Papua New Guinea displayed a 56% increase in processed food sales, with only 13.2% of their population residing in urban settings [58]. The small geographical size of some PICs may also render urbanisation less important.

The globalisation of food systems has meant that trade plays a substantial role in determining what foods are available to consumers in the Pacific, along with considerable growth in the market power of TNFBCs. The systematic reduction in barriers to the flow of goods, services and investments across borders achieved via bilateral, regional and multilateral trade agreements has contributed substantially to the Pacific’s nutrition transition. PICs that joined the World Trade Organization (WTO) have been required to change their trade policies to reduce tariff barriers to trade. Trade agreements, such as the Pacific Island Countries Trade Agreement (PICTA), which many PICs have ratified [59], also contain rules about how governments can regulate markets and may thereby limit the adoption of policy mechanisms designed to reduce consumption. This has led to not only greater availability and affordability of processed foods, but also a level of reliance on these foods as they displace traditional foods produced domestically [60,61]. 

There is evidence to show that the participation of PICs in WTO and free trade agreements has resulted in ‘dietary dependence’. Where prior to contact with colonial powers these countries were known to have ‘subsistence abundance’ [62], the saturation of imported products has reduced demand for locally produced foods [61,63]. Furthermore, the focus on exporting local foods as part of trade agreements means a reduction in the availability of these foods in the local food environment. When imported foods comprise a large share of the food supply, any factors that threaten that supply are a threat to food security. For the Pacific Islands, this can mean weather events, such as cyclones or storms, but it can also mean economic interferences, such as price increases or currency exchange [3]. The proportion of food expenditure on imported and nontraditional foods now approximates or exceeds 50% of total food expenditure in Kiribati and Tonga [61]. On top of this, imported foods now form up to 72% of Kiribati diets [61]. Given that human-made climate change is increasing the frequency of extreme weather in this region, this will have important implications for food security among PICs [62]. Furthermore, in Fiji, costs associated with producing fresh foods (traditional starchy staples, vegetables, and fruits) have led to fresh produce becoming less affordable for its population as it competes with imported foods [64]. 

We have demonstrated that the types of vegetable oils sold in Pacific markets differs between H-ICs and U- and L-MICs. Where olive oil is largely the staple oil in diets of Australians and New Zealanders, palm oil and “other” edible oils (such as coconut oil or blended vegetable oils) appear to play a much larger role in PIC imports and dietary intake. This is possibly a result of the WTO trade liberalisation that occurred during the mid-1990s, when palm oil from Malaysia and Indonesia began to take a much larger share of the edible oil market [65]. WHO notes that fat from vegetable oils has been added to existing levels of fat contributed by coconuts, i.e., these edible oils have not replaced traditional oils but added to them [12]. Whilst the evidence is inconclusive, palm oil is noted to have particular implications for nutrition and health, including increased risk for cardiovascular diseases [66], increased LDL cholesterol [67] and impaired glucose tolerance (i.e., insulin sensitivity) [68].

There are also policy factors that may be driving the observed trends. For example, many PIC countries have adopted policy and regulatory actions (and in some cases have been world-leaders in doing so) targeting processed food and beverage markets. This includes sugar-sweetened beverage (SSB) taxes in twelve countries [69], as well as import taxes on processed foods, subsidies towards fresh produce and point-of-entry import duties on foods high in sugar [44]. Some of these regulatory actions may explain why beverage sales are increasing at a lower rate than those of processed foods in U-MICs and L-MICs, and in the case of Vanuatu even declining. Of the countries included in this analysis, eight have incorporated SSB taxes as either excise taxes or import tariffs on sugary drinks, including American Samoa, French Polynesia, Fiji, Kiribati, Tonga, Samoa, Nauru and Vanuatu [44,69]. 

However, this existence of taxes does not help to explain some other observations noted in our analysis. For example, Nauru and Tonga introduced SSB taxes in 2007 and 2013, respectively, but increases in sales for carbonates were observed in both countries between 2004 and 2018. These observations may be a result of a lack of sensitivity in the dataset. The nuances of these taxes could also have effects on sales of other products, for example, juice. In 2015, Vanuatu introduced a $0.46 USD tax on carbonated beverages containing added sugar or other sweeteners, which may suggest that consumers have shifted towards juice as an alternative beverage choice. Moreover, countries like Fiji have recognised the need to change the ‘obesogenic’ environment through policy and regulations on TNFBCs but have faced immense pressure from industry in doing so [70,71]. Therefore, the role of the food industry in shaping policy and regulatory responses in the Pacific, and hence as a distinct food systems driver, must also be considered. The food industry and in particular transnational food and beverage companies operating in Fiji, for example, has engaged in a diversity of political activities to undermine policy responses and generate a favourable corporate image [71]. Similarly, in Australia, the food industry and associated peak lobby groups have engaged in policy substitution and extensive lobbying activities to undermine regulatory responses targeting unhealthy foods and obesity prevention [72]. The impact of various food taxes on sales and consumption is under-investigated. Qualitative evidence in Tonga suggests that food taxes have reduced sales of mutton flaps and turkey tales [73]. Future studies are needed to comprehensively assess the impact of food taxes on sales and consumption in the region.

### 4.2. Limitations

There are several limitations of this analysis, warranting caution in the interpretation of our findings. Euromonitor only collects data on formal sales channels, such as grocery stores and food service outlets, and there may therefore be gaps in the data where local, informal systems have not been captured. Furthermore, the data used in this analysis for a number of PICs were modelled by Euromonitor rather than collected within countries and are at best illustrative of general trends in the types of product categories sold rather than an actual representation. This is an important limitation, and the results should be interpreted with caution. Because of this limitation, the data are insensitive to abrupt changes resulting from policy change. We were therefore unable to ascertain changes resulting from the implementation of food and nutrition policy measures or regulations. Additionally, the market data capture sales volumes only and are not a measure of consumption. This is because they do not incorporate other potential sources of processed foods, including those made at home or sold via informal markets, nor do they account for food waste. Furthermore, the review of food systems drivers is not comprehensive. The paper does not describe changes in fresh food markets and hence does not provide a comparative assessment of the role of processed foods in the wider nutrition transition incorporating all food categories. This is a consideration for future studies. Furthermore, we were unable to disaggregate the Euromonitor data to apply the NOVA classification in the analysis, and therefore unable to determine changes in the proportion of UPFs in Pacific diets.

## 5. Conclusions

The findings of this study suggest that a processed food nutrition transition is well underway in the Pacific region, most prominently in L-MICs and U-MICs. Whilst sales of processed foods are plateauing and even decreasing in some H-ICs, they are increasing in L-MICs and U-MICs in the region, with potentially important implications for dietary quality and health. In particular, baked goods, edible oils (especially palm oil), processed meat, noodles and sweet biscuits are making up a considerable proportion of these sales. Sales of soft drinks had a less evident increase across all income groups, but in some countries, carbonated soft drinks outweighed sales of all other beverage categories by double. Furthermore, L-MIC and U-MIC countries that experienced slowing sales of carbonated soft drinks simultaneously saw increases in other sugary drinks, such as juice. These trends can partially be explained by several primary food system drivers, including income growth, urbanisation and the increasing availability of these food products as a result of globalised trade. In some PICs, policy decisions such as SSB taxes may also explain why soft drink market growth appears to be stagnating although our data are insensitive to such changes, and hence, further research is needed. As further evidence mounts on the adverse outcomes linked with ultra-processed foods in particular, dietary advice for Pacific populations is straightforward: Eat less ultra-processed foods and consume more whole or minimally processed foods.

## Figures and Tables

**Figure 1 nutrients-11-01328-f001:**
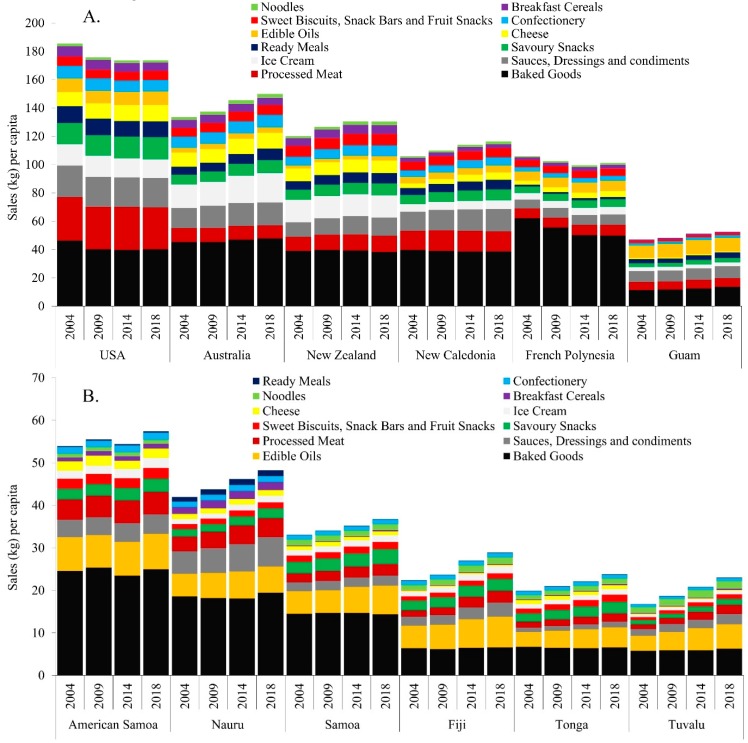
Per capita processed food sales in (**A**) high-income, (**B**) upper-middle income and (**C**) lower-middle income Pacific island countries, 2004–2018. Note: Different scales are used for each graph to demonstrate granularity in product categories; categories are ranked from highest (bottom) to lowest (top) volume and align with the same product category order in the figure keys.

**Figure 2 nutrients-11-01328-f002:**
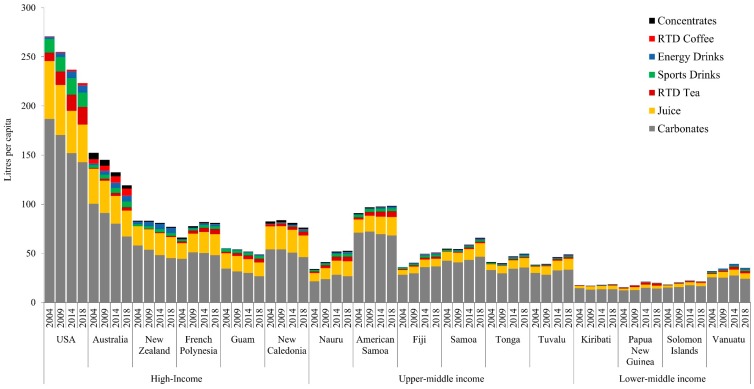
Per capita soft drink sales in Pacific island countries, 2004–2018, by country income level.

**Figure 3 nutrients-11-01328-f003:**
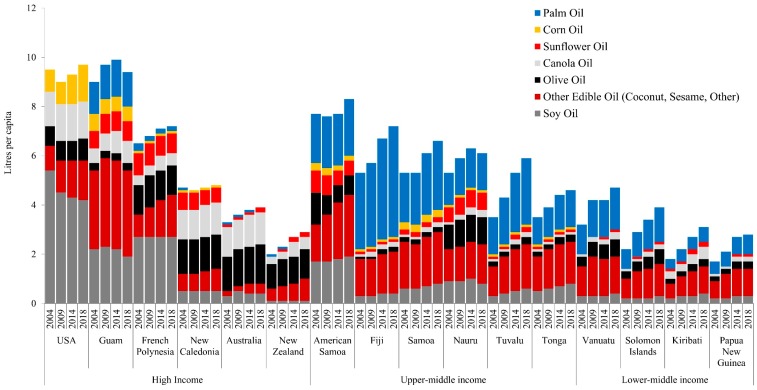
Per capita vegetable oil sales in Pacific island countries, 2004–2018, by country income level.

**Table 1 nutrients-11-01328-t001:** Included processed food product categories and descriptions.

Product Category	Description
Total processed foods	Aggregation of all processed food categories
Baked goods	Bread, pastries, dessert mixes, frozen baked goods, cakes
Breakfast cereals	Ready-to-eat and hot cereals
Confectionary	Chocolate confectionery, sugar confectionery, gum, chocolate spreads, jams and preserves, nut and seed-based spreads
Cheese	Processed and unprocessed cheeses.
Instant noodles	Instant noodles
Vegetable oils	Packaged liquid/fluid cooking oils made from seeds/fruits
Ice cream and frozen desserts	Ice cream, frozen desserts
Processed meat and seafood	Processed meat, meat substitutes, processed seafood
Ready meals	Shelf stable, frozen, dried, chilled ready meals, dinner mixes, frozen pizza, chilled pizza
Sauces, dressings and condiments	Pastes/purees, stock cubes, monosodium glutamate, pasta sauces, cooking sauces, ketchup, mayonnaise, salad dressings, others
Savoury snacks	Fruit snacks, chips/crisps, extruded snacks, tortilla/corn chips, popcorn, pretzels, nuts, other savoury snacks
Sweet biscuits, snack bars and fruit snacks	Sweet biscuits, savoury biscuits and crackers, and bread substitutes, granola/muesli bars, breakfast bars, energy and nutrition bars, fruit bars
Total beverages	Aggregation of all beverage categories
Carbonated soft drinks	Sweetened drinks containing dissolved CO_2_, regular and low calorie, naturally and artificially sweetened included.
Concentrates	Liquid concentrates and powder concentrates
Fruit and vegetable juice	100% juice, nectars (25–99% juice), juice drinks (<24%), flavoured drinks
RTD coffee, RTD tea	Packaged ready-to-drink (RTD) coffee, excluding coffee flavoured milk drinks, Still RTD tea, carbonated RTD tea
Sports and energy drinks	Sports and energy drinks
Asian specialty drinks	Traditional Asian, lactic acid and partial yoghurt drinks

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
