# Peer review of "Processed Foods and Nutrition Transition in the Pacific: Regional Trends, Patterns and Food System Drivers"

_nutrients, 2019, doi:10.3390/nu11061328_

Round 1

Reviewer 1 Report

A problem with this paper is the confusing use of frameworks for understanding changes in diet in this region. The NOVA classification is used when discussing health outcomes, but then the choice of foods to include in the analysis seems arbitrary and not in line with this classification. Choosing only one processed culinary ingredient for further analysis (why not sugar?) along with processed foods not related to health outcomes (e.g. cheese, yoghurts and sour milk - generally linked with protective effects against NCDs) as well as ultra-processed foods, is confusing.    

To justify this approach, the manuscript could clearify (by sectioning the introduction and discussion) that the nutrition transition involves at least two main changes; adding more foods ready to eat or heat (ultra-processed) at the expense of traditional foods, as well as changing the way people cook at home by the availability and affordability of cooking oils (processed culinary ingredients), and linking this with the contribution of ultra-processed foods and cooking oils, respectively.

Authors should consider leaving out processed foods not linked with health outcomes. 

Also, a dicussion of the general encouragement  of use of vegetable cooking oils in most dietary guidelines would be useful. The percetion that vegetable oils are "healthy", brought about both by the food industry and nutrition professionals, has likely contributed to the large increase in use. The contribution of energy from these highly promoted oils is substantial.  

Single points: 

57-58 and 68: Authors should consider summarizing these and adding more health outcomes, as several other diet-related NCDs have been linked with ultra-processed foods. Also, including Kevin Hall's recent RCT study on ultra-processed foods and short term weight gain could be informative. 

82-85: This section is unclear. 

104-105: Too conclusive? ...key literature to indentify potential drivers of the observed... 

Figure 1 A-C. It is not possible to distinguish between baked good and ready meals in the printed copy. 

264: The availability and affordability of processed foods also change in-house cooking, thereby changing traditional diets in at least two ways (matching the discussion on lines 237-243). 

Author Response

We kindly thank the reviewers for their thoughtful and helpful comments, which we believe have helped to strengthen the paper. We have responded to each comment as itemised in the following sections.

REVIEWER 1

Comment 1: A problem with this paper is the confusing use of frameworks for understanding changes in diet in this region. The NOVA classification is used when discussing health outcomes, but then the choice of foods to include in the analysis seems arbitrary and not in line with this classification. Choosing only one processed culinary ingredient for further analysis (why not sugar?) along with processed foods not related to health outcomes (e.g. cheese, yoghurts and sour milk - generally linked with protective effects against NCDs) as well as ultra-processed foods, is confusing.    

Response: We acknowledge these important points made by the reviewer and point to the following. First, in the introduction we state that we adopt a broad definition of processed foods that includes but is not limited to the ultra-processed category. This is because we intended to capture processed and culinary ingredients categories not included in the UPF category but important to the nutrition transition. In particular, this included vegetable oil which others (e.g. Barry Popkin) have noted as particularly important to the expanding calorie supply in low and middle income countries, and the implications it raises for nutrition.

However, the reviewer is correct, we were mistaken in failing to use the NOVA terms and now refer to ultra-processed foods, processed foods, and culinary ingredients in the introduction. Unfortunately the Euromonitor data does not allow for disaggregation of categories by NOVA classification, which also justifies the use of a broader definition of processed foods. We have further noted this in the limitations section of the paper.

We agree with the reviewer regarding not including yoghurt and sour milks in our analysis; including this was an error in Table 1 and this category does not appear in the results or figures, and we have now removed this category from Table 1.

Comment 2: To justify this approach, the manuscript could clarify (by sectioning the introduction and discussion) that the nutrition transition involves at least two main changes; adding more foods ready to eat or heat (ultra-processed) at the expense of traditional foods, as well as changing the way people cook at home by the availability and affordability of cooking oils (processed culinary ingredients), and linking this with the contribution of ultra-processed foods and cooking oils, respectively.

Response: We have included a statement in the introduction (lines 80-82) stating; “The nutrition transition not only involves a transition from more traditional to globalised and processed foods, but also substantial changes in the way people at home source and prepare food (40)”

This is further discussed in our Discussion section (lines 254-258). “The increase in vegetable oils and especially palm oil sales in many countries appears to be a noteworthy feature of the nutrition transition in the Pacific. Others have reported that as countries transition vegetable oils appear to markedly increase in availability and consumption, as a cheap and convenient culinary ingredient used in home cooking.”  

Comment 3: Authors should consider leaving out processed foods not linked with health outcomes. 

Response: The reviewer is correct that some of the processed food categories we include have not been linked to health outcomes. However, several are vital to food security and contribute substantially to the food supply in the region. We therefore include some basic processed food (e.g. breads within baked goods) and culinary ingredients (e.g. vegetable oils) categories as these have their own additional critical implications for nutrition in transitioning countries. The inclusion of vegetable oils also helps to illustrate important changes in food systems and culinary practices in the region.

Comment 4: Also, a discussion of the general encouragement of use of vegetable cooking oils in most dietary guidelines would be useful. The perception that vegetable oils are "healthy", brought about both by the food industry and nutrition professionals, has likely contributed to the large increase in use. The contribution of energy from these highly promoted oils is substantial.  

Response: One contributing factor to the dramatic increase in vegetable oil usage is that many food based dietary guidelines recommend limiting the intake of foods high in saturated fats and advise replacing these foods with small amounts of foods such as vegetable oils to achieve this recommendation. This advice has at times been misrepresented and created a perception among consumers that vegetable oils are ‘healthy’ and to be consumed in large amounts despite their high energy density. The contribution of vegetable oils to the supply of total calories in transitioning countries has been well described (e.g. Popkin), however there appears to be limited evidence on the health implications of specific vegetable oils or of vegetable oils in general. However, there is evidence that dietary guidelines are recognising the risks posed by vegetable oils and are advising limited consumption; see, for example, Fiji’s dietary guidelines: http://www.fao.org/3/a-as883e.pdf. We have added this lines 250-252.

Comment 5: 57-58 and 68: Authors should consider summarizing these and adding more health outcomes, as several other diet-related NCDs have been linked with ultra-processed foods. Also, including Kevin Hall's recent RCT study on ultra-processed foods and short term weight gain could be informative.

Response: Thank you for your suggestion. We have included the suggested reference, and have updated this section in lines 59-65 “A growing body of evidence links a higher proportion of UPFs in the diet with obesity, cancer, cardiovascular disease, metabolic risks and all-cause mortality (23-26).”

Comment 6: 82-85: This section is unclear. 

Response: We have revised the sentence to say; “At the same time, because these food systems forces have varying and context-dependent impacts on the availability, affordability and desirability of different types of food, there is no singular or uniform nutrition transition” (Lines 85-87)

Comment 7: 104-105: Too conclusive? ...key literature to identify potential drivers of the observed... 

Response: Thank you, we have revised the sentence to the reviewer’s suggestion (now line 109).

Comment 8: Figure 1 A-C. It is not possible to distinguish between baked good and ready meals in the printed copy. 

Response: The categories are ranked from highest to lowest quantity and are aligned with the order of product categories in the figure key. To make this clearer to the reader we have added this guidance as a footnote immediately below Figure 1 A-C.

Comment 9: 264: The availability and affordability of processed foods also change in-house cooking, thereby changing traditional diets in at least two ways (matching the discussion on lines 237-243). 

Response: Thank you for your comment. We have included your suggestion in the discussion (lines 253-257).

Reviewer 2 Report

Summary

This article provides a well written and novel exploration of processed food trends in Pacific countries, with a descriptive analysis of the available Euromonitor data, and a discussion of the drivers of the nutrition transition in the Pacific. I thought the information about the types of processed foods that play the greatest role in the Pacific was really interesting. However, I had some concerns and questions about the limitations of the modeled data.

Major

A major problem with the Euromonitor data is that it is, as the authors describe, modeled for all 13 of the Pacific Island Countries (PICs) (excl NZ, Australia and US), using comparator countries and economic and demographic variables such as GDP, income inequality, life expectancy and consumer expenditure. GDP however is also is used to classify countries into HIC, HMIC, LMICs. As a result if the modeled sales trends and country classification both depend on GDP then they would be expected to be correlated, which is this study’s major finding. The authors’ conclusions about the food transition being most prominent in LMICs and UMICs, may be purely because of an assumption from the model rather than real data. Is it not still possible that high income Pacific countries are increasing in processed food sales, and that some LMICs and UMICs have declines in processed food sales but that these were not picked up by Euromonitor modelling?

Similarly, was there any evidence that the modelling by Euromonitor were sensitive enough to detect potential changes in processed food sales volumes after policy changes like food and beverage taxes?  Please expand more on what the impact of the Euromonitor modelling may mean, ie in the limitations. Statements made about policy impacts must be softened if the data were not designed to detect any true sudden changes. For example, Tonga import data demonstrates a plateauing in sweetened beverage import volumes 2009 to 2018 possibly associated with SSB taxes but there was no evidence of this in the study's figures. Another example, PNG has seen a rapid increase in sweetened beverage imports in recent years but this is not obvious in Figure 2.

-          How strong are Euromonitor networks in these Pacific countries to test and confirm the modelling results?

-          Is there further information on what comparator countries were used for these Pacific countries? Were any of these from the Pacific?

-          Does Pacific import data feed into modelling results?

-          Does Pacific local manufacturing data feed into modelling results?

Minor

I noticed Fiji was missed out from the list of countries with SSB taxes. I have thought they did have some sort of tax on SSBs? Has this changed?

Have food taxes influenced any of the author’s findings, eg fat and sugary food taxes in Tonga.

Author Response

We kindly thank the reviewers for their thoughtful and helpful comments, which we believe have helped to strengthen the paper. We have responded to each comment as itemised in the following sections.

Reviewer 2

Comment 1: A major problem with the Euromonitor data is that it is, as the authors describe, modeled for all 13 of the Pacific Island Countries (PICs) (excl NZ, Australia and US), using comparator countries and economic and demographic variables such as GDP, income inequality, life expectancy and consumer expenditure. GDP however is also is used to classify countries into HIC, HMIC, LMICs. As a result if the modeled sales trends and country classification both depend on GDP then they would be expected to be correlated, which is this study’s major finding. The authors’ conclusions about the food transition being most prominent in LMICs and UMICs, may be purely because of an assumption from the model rather than real data. Is it not still possible that high income Pacific countries are increasing in processed food sales, and that some LMICs and UMICs have declines in processed food sales but that these were not picked up by Euromonitor modelling?

Response: These are important data limitations identified by the reviewer that we attempted to address in both the methods section and limitations section in the discussion. Acknowledging these points, we have taken further steps to further note and clarify these limitations, including the specific point about the dependence of the modelled data on GDP identified by the reviewer (lines 339-343). We hope that these strong indications of the limitations of the data are adequate. Nonetheless, we also emphasise that in the absence of alternative sources of country-level data for Pacific Island countries, the modelled data provides best available estimates and insights for understanding the role of processed foods in the region’s nutrition transition.

Comment 2: Similarly, was there any evidence that the modelling by Euromonitor were sensitive enough to detect potential changes in processed food sales volumes after policy changes like food and beverage taxes?  Please expand more on what the impact of the Euromonitor modelling may mean, ie in the limitations. Statements made about policy impacts must be softened if the data were not designed to detect any true sudden changes. For example, Tonga import data demonstrates a plateauing in sweetened beverage import volumes 2009 to 2018 possibly associated with SSB taxes but there was no evidence of this in the study's figures. Another example, PNG has seen a rapid increase in sweetened beverage imports in recent years but this is not obvious in Figure 2. How strong are Euromonitor networks in these Pacific countries to test and confirm the modelling results? Is there further information on what comparator countries were used for these Pacific countries? Were any of these from the Pacific? Does Pacific import data feed into modelling results? Does Pacific local manufacturing data feed into modelling results?

Response: These are important points made by the reviewer. To clarify these points we followed-up with Euromonitor, seeking more information on their approach and specific method used to model the data and its sensitivity to aberrations. (We have broken down the above comment into separate questions, and addressed them accordingly).

Comment 3: Was there any evidence that the modelling by Euromonitor were sensitive enough to detect potential changes in processed food sales volumes after policy changes like food and beverage taxes? 

Response: As this is modelled data, it is unlikely that the data is sensitive enough to detect potential changes in processed food sales volumes as a result of policy changes. We have included this disclaimer in line 318-319.

Comment 4: Please expand more on what the impact of the Euromonitor modelling may mean, i.e. in the limitations. Statements made about policy impacts must be softened if the data were not designed to detect any true sudden changes. For example, Tonga import data demonstrates a plateauing in sweetened beverage import volumes 2009 to 2018 possibly associated with SSB taxes but there was no evidence of this in the study's figures. Another example, PNG has seen a rapid increase in sweetened beverage imports in recent years but this is not obvious in Figure 2.

Response: We agree with the reviewer that it is important to identify the limitations of our current data set, and have further done so in lines 339-343, including a statement regarding the sensitivity of the data to changes in policy. We acknowledge the reviewers last point, although note difficulties in comparing import data (as in the example provided) with the sales data we analysed in this paper.

Comment 5: How strong are Euromonitor networks in these Pacific countries to test and confirm the modelling results? Is there further information on what comparator countries were used for these Pacific countries? Were any of these from the Pacific? Does Pacific import data feed into modelling results? Does Pacific local manufacturing data feed into modelling results?

Response: We have added further information on how Euromonitor undertakes its modelling in the methods section (lines 135-146). The first step in the modelling process is to select a set (typically three or four) of similar researched countries for each modelled country, based on similarities in terms of economic, geographical, cultural, and industry related dimensions. Due to the lack of regional data, examples of comparitor countries used were Ecuador, Malaysia, Indonesia, Cambodia, Laos, Peru and Bangladesh. The set of available indicators to measure similarity is selected in advance, but the relative weighting for each indicator group in the final similarity index is determined during the calibration process described below. When constructing this similarity index, they assess some 50 indicators including GDP per capita and real GDP growth rate, urbanization rate, average household size, number of children, birth rates, age distribution indicators, infant mortality, life expectancy, adult literacy rate, corruption perceptions indices, female employment rates, religion and ethnicity indicators, climate indicators, trade intensity and geographical distance between countries. They then compare similarities for all of these indicators across every possible pairing of countries. Euromonitor Industry informants then review and sense check the final results and check the data for outliers. Import and local manufacturing data do not feed into these results as far as we were able to find. We were not able to ascertain the strength of Euromonitor networks in the region.

Comment 6: I noticed Fiji was missed out from the list of countries with SSB taxes. I have thought they did have some sort of tax on SSBs? Has this changed?

Response: We thank the reviewer for noting this omission. We have updated the text to include Fiji among the countries who have implemented a SSB tax.

Comment 7: Have food taxes influenced any of the author’s findings, eg fat and sugary food taxes in Tonga.

Response: We were unable to ascertain from the data the impacts of food taxes, as this was outside of the study design and arguably not possible given the data limitations noted earlier. Hence we have declined to elaborate on describing the impacts of specific taxes. We did search for further literature on this, finding that qualitative evidence in Tonga suggests food taxes had reduced sales of mutton flaps and turkey tails [http://www.fao.org/3/a-i8052e.pdf].This is evidently a research gap, and we have acknowledged this in lines 321-322.

Round 2

Reviewer 2 Report

The authors have adequately addressed my earlier comments. I have no further suggestions to improve the manuscript.